# Different trajectories of adolescent mental health problems before and over the course of COVID-19: Evidence of increase, decrease, and stability

Coriena de Heer◯*, Catrin Finkenauer, Gonneke Stevens◯

Department of Interdisciplinary Social Science, Utrecht University, Utrecht, The Netherlands

* c.a.deheer@uu.nl

## Abstract

The COVID-19 pandemic and related measures to control the spread of the virus have negatively impacted adolescent mental health. However, the pandemic may have been more consequential for the mental health of some adolescents than others. Little is known about the heterogeneity in mental health responses to the pandemic among adolescents. This study aims to identify and characterize different trajectories of adolescent mental health problems before and over the course of the COVID-19 pandemic. We used data from 1,521 adolescents ($M_{age}$ = 17.91) collected at four measurement points: autumn 2019 (pre-COVID-19), spring 2020, autumn 2020, and autumn 2021. Mental health problems were assessed using four indicators: emotional symptoms, conduct problems, hyperactivity-inattention problems, and peer relationship problems. Latent class growth analyses identified stable low, stable high, increasing, and decreasing trajectories for emotional symptoms, conduct problems, and hyperactivity-inattention problems, and a stable low and stable high trajectory for peer relationship problems. Adolescents with high mental health problems before and during the pandemic reported relatively low levels of family and friend support. Gender, migration background, and family socioeconomic status were not consistently associated with the trajectories across mental health problems. Our findings highlight the diverse impact of the COVID-19 pandemic on mental health problems among adolescents. This suggests that interventions and support strategies for adolescents to cope with stressful circumstances should be tailored to the specific needs of different groups of adolescents.

## Introduction

The COVID-19 pandemic and related measures to control spreading of the virus have increased concerns about adolescent mental health. Interactions with peers and access to formal mental health services within school, both of which are important

**Data availability statement:** The dataset and syntax are available through the Open Science Framework (OSF) repository (https://osf.io/j2t6w/).

**Funding:** This work was supported by the IMPROVA Project, funded by the European Union's Horizon Europe Research and Innovation Programme (Grant Agreement No. 101080934; https://www.improva-project.eu/). The funding was received by CH, CF, and GS. The funder had no role in study design, data collection and analysis, decision to publish, or preparation of the manuscript.

**Competing interests:** The authors have declared that no competing interests exist.

for adolescent mental health [1,2], were significantly disrupted due to lockdowns, school closings, and social distancing measures. Reviews and longitudinal studies suggest that the pandemic has negatively impacted adolescent mental health [3,4]. Dutch studies have also reported a decline in adolescent mental health during the pandemic [5,6]. Although these studies provide insight into adolescent mental health before and during the pandemic, most report average changes in mental health over time. However, it is likely that the pandemic did not have the same impact on all adolescents.

Theoretical frameworks, such as the differential susceptibility theory and diathesis stress model, suggest that individuals vary in their adaptation and reactivity to stressful events [7]. The pandemic might represent a long-lasting stressful event and, according to these frameworks, the pandemic and related measures could have been more consequential for the mental health of some adolescents than others. Therefore, this study aimed to investigate the heterogeneity in adolescent mental health trajectories before and over the course of the pandemic and characterize potential differences in such trajectories.

### Trajectories of adolescent mental health problems

Few studies have examined the heterogeneity in mental health responses to the COVID-19 pandemic among adolescents. Two studies in the UK, using data from the Co-SPACE study including over 3,000 children and adolescents, investigated mental health trajectories. Raw and colleagues [8] analyzed data from the first four months of the pandemic, and Guzman Holst and colleagues [9] analyzed data from the first 13 months. Despite some differences in the trajectories between the studies, both found multiple trajectories for emotional symptoms, conduct problems, and hyperactivity-inattention problems. Most children and adolescents exhibited stable low levels of mental health problems throughout the study. Additionally, increasing, decreasing, and stable high trajectories were identified. Moreover, Guzman Holst and colleagues [9] found a trajectory for emotional symptoms that initially increased and then decreased after seven months. In China, Wang and colleagues [10] measured anxiety and depressive symptoms among 2,352 adolescents three times from February 2020 to June 2020. They identified two trajectories: Most adolescents showed a low-level trajectory that slightly increased, while the other group showed a high-level trajectory that decreased over time. A limitation of the studies of Raw and colleagues [8] and Wang and colleagues [10] is that adolescents were monitored during a relatively short time period that only covered the beginning of the pandemic. In addition, all three studies failed to assess adolescent mental health *before* the pandemic, which prevents conclusions regarding the impact of the pandemic on adolescent mental health trajectories.

A Dutch study addressed the aforementioned limitations by measuring depression and anxiety symptoms in 188 adolescents both before the pandemic and at eight, ten, and fifteen months after its onset in the Netherlands [11]. The study revealed three trajectories: Most adolescents showed a stable low-level or stable moderate-level trajectory, while a small subgroup showed a high-level trajectory that decreased during the

pandemic. However, the sample size of this study was small, and the group of adolescents in the decreasing high-level trajectory only consisted of five adolescents. Small sample sizes reduce the robustness and generalizability of identified trajectories because they lack statistical power to detect these trajectories and accurately represent the diversity of the population.

This longitudinal study investigated the heterogeneity of adolescent mental health trajectories over the course of the pandemic. Addressing the limitations of previous research, this study assessed mental health problems in a large sample of adolescents before the pandemic and three times during the first 18 months of the pandemic. Examining trajectories over a longer time period provides a deeper understanding of how adolescent mental health has developed during this critical period and therefore offers more valuable insights for interventions targeting mental health problems than short-term studies. This deeper understanding is necessary to determine the necessity of interventions to maximize their impact and be tailored to adolescents' needs, circumstances, and readiness to engage in these interventions. For example, adolescents go through significant changes, and some issues may be developmentally normal or typical in the context of a national pandemic (e.g., anxiety). Longitudinal data allow us to examine whether behaviors and feelings persist or worsen over time, which can help improve the effectiveness of interventions. Moreover, the study assessed both indicators of internalizing and externalizing problems to provide a more comprehensive understanding of trajectories of mental health problems in adolescence.

### Predictors of adolescent mental health problems trajectories

To identify adolescents who may benefit most from prevention and intervention efforts, it is necessary to consider potential predictors of adverse trajectories. Several demographic and social characteristics may be related to the different trajectories of adolescent mental health problems before and during the pandemic.

First, research suggests that some groups of adolescents are generally more vulnerable to mental health problems, such as adolescents with a migration background or from families with lower socioeconomic status (SES) [12,13]. Having a migration background or growing up in lower SES families is related to stressful experiences such as social exclusion, discrimination, and precarious living conditions [14,15]. The pandemic may have exacerbated the already stressful daily experiences and increased the vulnerability to mental health problems among adolescents with a migration background or from lower SES families. A German study provided evidence for this suggestion, indicating that these adolescents showed a larger increase in mental health problems during the pandemic than adolescents without a migration background or from higher SES families [16]. However, a Dutch study found little evidence of associations between the course of adolescent mental health during the pandemic and migration background or family SES [6].

Gender may also play a role. Research has shown that boys tend to report more externalizing problems, while girls tend to report more internalizing problems [17]. Moreover, during the pandemic, girls showed greater increases in internalizing problems than boys [18], suggesting that the pandemic may have increased gender disparities in mental health problems. However, another study found no gender differences in changes in mental health indicators during the pandemic [6].

In addition to demographic variables the presence of family and friend support may predict trajectories of mental health problems. Social support is related to lower levels of mental health problems and can buffer the negative effects of stressful situations on mental health [19]. During the pandemic, social support can protect adolescents from experiencing mental health problems [20,21]. Additionally, adolescents who reported lower social support in general or from their families were more likely to be in trajectories with more mental health problems than in trajectories with stable low mental health problems [9–11]. While the role of family support in adolescent mental health during the pandemic has been well-researched, the role of friend support has received less attention in the literature.

### Current study

First, we aimed to identify different trajectories of adolescent mental health problems before and during the COVID-19 pandemic, using a sizeable, prospective, longitudinal study. Mental health problems were assessed by four indicators: emotional symptoms, conduct problems, hyperactivity-inattention problems, and peer relationship problems. Previous

studies found different trajectories of adolescent mental health problems during the pandemic, including trajectories reflecting improvement, deterioration, and high/low stability. We expected to find similar trajectories. Second, we aimed to explore whether migration background, family SES, gender, family support, and friend support were associated with the identified trajectories. Based on the limited research available, we cautiously hypothesized that adolescents with a migration background, from lower SES families, and who received less support from family and friends might be overrepresented in stable high or deteriorating mental health problems trajectories. Additionally, we predicted that girls would be overrepresented in the stable high internalizing problems trajectories and boys would be overrepresented in the stable high externalizing problems trajectories.

## Method

### Participants and procedure

We used data from the YOUth Got Talent project, a longitudinal study focusing on the well-being of Dutch adolescents. Adolescents in three tertiary vocational schools in Utrecht filled out a questionnaire. In total, 1,602 adolescents participated at least once out of the four measurements. This sample size was determined based on practical considerations, specifically the number of schools, school departments, and adolescents willing to participate in the study. After excluding participants due to non-serious responses ($n = 44$), missing data on the SDQ-R across waves ($n = 36$), and an age outlier (36.21 years; $n = 1$), 1,521 adolescents remained eligible for analysis. The first measurement was conducted pre-COVID-19 in autumn 2019 (T1, 27 September 2019 to 14 February 2020, $n = 1,209$). The second to fourth measurements took place during the pandemic: in spring 2020 (T2, 7 May 2020 to 25 June 2020, $n = 823$), autumn 2020 (T3, 26 October 2020 to 8 January 2021, $n = 574$), and autumn 2021 (T4, 25 October 2021 to 28 January 2022, $n = 575$). The mean time interval between T1 and T2 was 6.56 months ($SD = 1.15$), between T2 and T3 was 5.10 months ($SD = 0.77$), and between T3 and T4 was 12.64 months ($SD = 1.42$). Participation rates from T1 to T4 were 79.5%, 54.1%, 37.7%, and 37.8%, respectively, when compared to the total sample. Attrition was mostly due to the fact that entire classes or an entire school dropped out in specific waves. Other reasons for attrition were that individual adolescents dropped out of school, finalized their education, or were absent during the questionnaire administration. The mean age of the participants at T1 was 17.91 years ($SD = 2.05$, range: 15.12-31.04), 41.0% were boys, and 25.3% had a migration background. Participants were considered to have a migration background if at least one parent was born outside the Netherlands.

Prior to each measurement, researchers informed adolescents about the purpose of the project, and that participation was voluntary and anonymous. Before filling out the questionnaire online, the adolescents provided written active consent. Researchers administered the self-report questionnaires in the classroom (T1, T4) or during online lessons (T2, T3). The questionnaires took about 20–30 minutes to complete. The Ethics Assessment Committee of the Faculty of Social and Behavioural Sciences at Utrecht University provided ethical approval (FETC18-070) in 2018 for the initial project and in 2020 and 2021 for updates of the project.

### Measures

**Mental health problems.** Mental health problems were measured at all waves using the revised version of the Strengths and Difficulties Questionnaire (SDQ-R) [22,23]. The SDQ-R has better psychometric properties than the original self-report SDQ [23], and asks about feelings and behaviors over the past six months. Answer categories were *not true* (0), *somewhat true* (1), and *certainly true* (2). The SDQ-R consists of four subscales and in total 15 items: emotional symptoms (5 items, e.g., I am often unhappy, down-hearted or tearful), conduct problems (4 items, e.g., I get very angry and often lose my temper), hyperactivity-inattention problems (3 items, e.g., I am restless, I cannot stay still for long), and peer relationship problems (3 items, e.g., other children or young people pick on me or bully me). We calculated mean scores for participants who completed more than half of the items. To retain comparability with the original SDQ, we multiplied the mean scores by five because the SDQ consists of 5-item subscales, and higher scores indicate higher

problem levels (ranging from 0 to 10). The emotional symptoms (ordinal α = .81-.84) and hyperactivity-inattention problems (ordinal α = .79-.81) subscales exhibited satisfactory internal consistency, whereas the conduct problems (ordinal α = .58-.71) and peer relationship problems (ordinal α = .51-.59) subscales exhibited low internal consistency, consistent with previous research [23].

To interpret the scores obtained from the SDQ-R, we applied cutoff values based on normative data for Dutch 17-year-old adolescents [24]. These cutoffs categorize scores into three groups: 'normal', 'borderline', and 'abnormal'. Adolescents with scores at or above the 90th percentile were classified as 'abnormal', representing those with the most extreme scores. Adolescents scoring in the next 10%, corresponding to the 80th percentile, were categorized as 'borderline'. For emotional symptoms, scores of six or higher were considered 'abnormal', and five 'borderline'. For conduct problems, scores of three or higher were 'abnormal', and two 'borderline'. For hyperactivity-inattention problems, scores of seven or higher were 'abnormal', and six 'borderline'. For peer relationship problems, scores of four or higher were 'abnormal', and three 'borderline'.

**Demographics.** Demographic information was assessed at T1 (or T2/T3/T4 if the participant did not participate at T1), which included gender (0 = *girl*, 1 = *boy*), migration background (0 = *both parents born in the Netherlands*, 1 = *at least one parent born outside the Netherlands*), and family SES. Family SES was measured using the 6-item Family Affluence Scale (FAS), which assesses material assets in the household (e.g., number of cars) [25]. The FAS is a reliable and valid instrument and correlates with other SES indicators, such as parent-reported income [25].

**Social support.** Family and friend support were measured at T1 by the Multidimensional Scale of Perceived Social Support (MSPSS) [26]. The MSPSS consists of two subscales: family support (4 items, e.g., my family really tries to help me) and friend support (4 items, e.g., I can talk about my problems with my friends). Items were rated on a 7-point Likert scale from 1 (*strongly disagree*) to 7 (*strongly agree*), so higher scores reflected more family or friend support ($\alpha_{family}$ = .92, $\alpha_{friends}$ = .92).

## Analyses

To investigate adolescent mental health trajectories before and during the pandemic, we conducted latent class growth analysis (LCGA) using Mplus version 8.8. LCGA is useful for identifying subgroups (i.e., classes) with different developmental trajectories within the larger population [27]. We selected LCGA over latent growth mixture modeling because our primary interest was to identify subgroups with similar patterns, rather than to model individual differences within those subgroups. In LGCA models, variances of the latent growth factors (i.e., intercepts and slopes) are constrained to zero within classes, which imposes that individuals within a class have similar trajectories. LCGA were conducted for each subscale of the SDQ-R separately.

First, we specified single-class growth models. The intervals between time points were adjusted to correspond with the time period between the measurements (i.e., the time points for the linear slope factor were fixed at 0, 0.5, 1, and 2). We used starting values 100 for the single-class growth models. The Satorra-Bentler scaled chi-square difference test was executed to test whether a linear or a quadratic growth model better fitted the data. If the slope variances were negative but non-significant, they were constrained to zero. Second, we specified LCGA models to determine the number of classes. The number of classes was established based on a trade-off between fit indices, parsimony, and interpretability [27]. The best fitting model had the lowest Bayesian Information Criteria (BIC) and a significant improvement in model fit compared to a model with one class less based on the Lo-Mendell-Rubin adjusted Likelihood Ratio Test (LMR-LRT) and Bootstrap Likelihood Ratio Test (BLRT). Additionally, the classes needed to have a sufficiently large size (*n* > 50) and be substantively interpretable as distinguishable classes. We used starting values 100 10 for these models. If we encountered computation problems, we increased the LRTSTARTS to 500 100 250 50. To avoid convergence issues, the residual variances and the variance-covariance matrix were fixed across latent classes.

After estimating the best fitting model and establishing the number of classes, we used a three-step approach to examine the extent to which the identified classes differed from each other on the demographic and social support variables [27]. First, the latent class model was estimated. Second, the most likely class variable was created using the latent class posterior distribution. Third, we added the demographic and social support variables one by one and tested the equality of means or percentages across classes using chi-square tests. In this third step, misclassification from the second step was taken into account.

Little's chi-square test for missing data indicated that the missing data on the mental health problems subscales were not completely missing at random, $\chi^2$ (144) = 191.52, $p = .005$. Adolescents who participated in all waves reported more emotional symptoms at T1 than adolescents who did not participate in all waves ($M = 3.43$ vs. $M = 3.00$), and less often had a migration background (15% vs. 27%). We did not find differences in conduct problems, hyperactivity-inattention problems or peer relationship problems at T1, gender and family SES between adolescents who participated in all waves and those who did not. To handle missing data without deleting cases and to limit the bias associated with missing data, as well as to account for the non-normality of the outcome variables, we used maximum likelihood with robust standard errors (MLR) [28]. We followed the Guidelines for Reporting on Latent Trajectory Studies (GRoLTS) checklist (S1 Checklist) [29] and Strengthening the Reporting of Observational Studies in Epidemiology (STROBE) statement (S2 Checklist) [30]. The dataset and syntax are available on the Open Science Framework (OSF, https://osf.io/j2t6w/). Additionally, plots displaying the estimated mean trajectories for each model, as well as plots showing the estimated means of the final model and the observed individual trajectories for each latent class, are available on OSF.

## Results

### Descriptive statistics

Table 1 shows the descriptive statistics and correlations for all study variables at the different measurement points. Girls reported more emotional symptoms than boys. Adolescents with a migration background reported more conduct problems than adolescents without such a background, and adolescents from lower SES families reported higher levels of emotional symptoms and peer relationship problems. Furthermore, more family support was related to lower levels of emotional symptoms, conduct problems, hyperactivity-inattention problems, and peer relationship problems. More friend support was related to lower levels of emotional symptoms and peer relationship problems. All four indicators of mental health problems were positively associated with each other within waves.

### Emotional symptoms trajectories

Emotional symptoms showed a nonlinear course over the four waves, as indicated by the significant chi-square difference test, suggesting that the quadratic growth model fitted the data better than the linear growth model, $\Delta\chi^2$ (1) =17.45, $p < .001$. The two- to four-class models showed relatively good fit as indicated by lower BIC values, and significant LMR-LRT and BLRT values ($p < .05$, Table 2), suggesting that a model with more classes fits the data better. A five-class model did not improve the model according to the LMR-LRT ($p = .622$). Compared to the four-class model, the five-class model separated one class into two classes that shared similar patterns. Thus, based on the fit indices and the criterion that classes should be substantively interpretable as distinguishable classes, the four-class model was selected for further analyses.

Fig 1 depicts the four trajectories of adolescents' emotional symptoms. Almost half of the adolescents fell into the stable low-risk group (46%), characterized by minimal emotional symptoms before and during the pandemic (Table 3). The increasing low-risk group (16%) showed low levels of emotional symptoms before the pandemic but an increase during the pandemic that leveled off over time. The decreasing moderate-risk group (24%) consisted of adolescents with moderate levels of emotional symptoms before the pandemic that decreased during the pandemic. Emotional symptoms in

**Table 1. Descriptive statistics and correlations for study variables.**

| | | n | Mean | SD | 1 | 2 | 3 | 4 | 5 | 6 | 7 | 8 |
|---|---|---|---|---|---|---|---|---|---|---|---|---|
| | 1. Gender[a] | 1,518 | | | | | | | | | | |
| | 2. Migration background[b] | 1,492 | | | −.03 | | | | | | | |
| | 3. Family SES (T1) | 1,195 | 8.64 | 1.80 | .07* | −.22** | | | | | | |
| | 4. Family support (T1) | 1,177 | 5.82 | 1.39 | −.01 | .00 | .11** | | | | | |
| | 5. Friend support (T1) | 1,179 | 5.88 | 1.23 | −.04 | −.03 | .05 | .34** | | | | |
| T1 | 6. Emotional symptoms | 1,192 | 3.08 | 2.53 | −.29** | −.11** | −.10** | −.32** | −.21** | | | |
| | 7. Conduct problems | 1,191 | 0.94 | 1.35 | .09** | −.01 | −.01 | −.23** | −.13** | .25** | | |
| | 8. Hyperactivity-inattention problems | 1,193 | 4.46 | 3.02 | −.03 | −.13** | .09** | −.17** | −.02 | .32** | .28** | |
| | 9. Peer relationship problems | 1,192 | 2.61 | 1.94 | −.04 | .00 | −.05 | −.21** | −.28** | .36** | .21** | .11** |
| T2 | 6. Emotional symptoms | 805 | 3.54 | 2.55 | −.32** | −.02 | −.12** | −.26** | −.15** | | | |
| | 7. Conduct problems | 806 | 1.03 | 1.49 | .03 | .10** | −.04 | −.20** | −.08* | .27** | | |
| | 8. Hyperactivity-inattention problems | 806 | 5.07 | 3.04 | −.03 | −.08* | −.01 | −.15** | .00 | .33** | .29** | |
| | 9. Peer relationship problems | 806 | 2.84 | 1.96 | −.05 | .01 | −.08* | −.23** | −.25** | .31** | .24** | .07* |
| T3 | 6. Emotional symptoms | 558 | 3.67 | 2.66 | −.36** | .00 | −.15** | −.20** | −.13** | | | |
| | 7. Conduct problems | 558 | 1.06 | 1.43 | .09* | .10* | −.04 | −.15** | −.09 | .25** | | |
| | 8. Hyperactivity-inattention problems | 559 | 5.41 | 3.01 | −.06 | −.02 | .04 | −.14** | −.02 | .35** | .22** | |
| | 9. Peer relationship problems | 558 | 2.72 | 1.93 | −.05 | .07 | −.13** | −.15** | −.30** | .33** | .24** | .05 |
| T4 | 6. Emotional symptoms | 560 | 3.65 | 2.57 | −.25** | −.05 | −.13* | −.20** | −.16** | | | |
| | 7. Conduct problems | 560 | 0.92 | 1.30 | .03 | .16** | .00 | −.13* | −.07 | .26** | | |
| | 8. Hyperactivity-inattention problems | 560 | 4.94 | 2.99 | −.08 | −.08 | −.01 | −.17** | .02 | .39** | .30** | |
| | 9. Peer relationship problems | 560 | 2.72 | 1.96 | −.04 | .04 | −.20** | −.12* | −.23** | .26** | .15** | .09* |

*Note.* Pearson correlations for the main study variables are shown at each time point. Point-biserial correlations were conducted for gender and migration background. SES = Socioeconomic status.

[a]0 = *girl* and 1 = *boy*.

[b]0 = *no migration background* and 1 = *migration background*.

*p < .05.

**p < .01.

both the increasing-risk and decreasing-risk groups remained within the 'normal' range. Finally, the stable high-risk group (14%) reported high levels of emotional symptoms before and during the pandemic, which were in the 'abnormal' range.

Adolescents in the increasing low-risk group did not differ from those in the stable low-risk group, except for a higher proportion of girls. Adolescents in the decreasing moderate-risk and stable high-risk groups were also more likely to be girls. Additionally, adolescents in both trajectories were less likely to have a migration background and perceived less family and friend support than adolescents in the stable low-risk group. Moreover, adolescents in the stable high-risk group had a lower family SES than adolescents in the stable low-risk group.

## Conduct problems trajectories

Similar to emotional symptoms, the quadratic growth model showed a better fit for conduct problems than the linear growth model, $\Delta\chi^2$ (1) =7.29, *p* = .007. The two- to four-class models of conduct problems showed a relatively good fit as indicated by lower BIC and significant BLRT values (*p* < .001, Table 2), suggesting that a model with more classes fits the data better. A three-class model did not improve the model fit over a two-class model according to the LMR-LRT (*p* = .212). However, a four-class model yielded a marginally significant LMR-LRT value, suggesting that a four-class model may fit the data better than the three-class model (*p* = .053). Moreover, the four-class model separated one class in the

**Table 2. Model fit indices of the LCGA models for the mental health indicators.**

| | Classes | BIC | Entropy | LMR-LRT *p*-value | BLRT *p*-value | Minimum class size |
|---|---|---|---|---|---|---|
| Emotional symptoms | 1 | 13,599 | | | | 1,520 |
| | 2 | 13,760 | .794 | <.001 | <.001 | 372 |
| | 3 | 13,427 | .728 | <.001 | <.001 | 246 |
| | 4 | 13,383 | .648 | .046 | <.001 | 210 |
| | 5 | 13,365 | .631 | .622 | <.001 | 142 |
| Conduct problems | 1 | 10,452 | | | | 1,520 |
| | 2 | 10,159 | .861 | .010 | <.001 | 175 |
| | 3 | 9,899 | .847 | .212 | <.001 | 62 |
| | 4 | 9,744 | .793 | .053 | <.001 | 63 |
| | 5 | 8,573 | .879 | .624 | <.001 | 27 |
| Hyperactivity-inattention problems | 1 | 14,722 | | | | 1,521 |
| | 2 | 14,934 | .686 | <.001 | <.001 | 678 |
| | 3 | 14,671 | .670 | <.001 | <.001 | 408 |
| | 4 | 14,615 | .635 | .006 | <.001 | 285 |
| | 5 | 14,574 | .649 | .003 | <.001 | 128 |
| | 6 | 14,569 | .607 | .273 | <.001 | 142 |
| Peer relationship problems | 1 | 12,437 | | | | 1,521 |
| | 2 | 12,544 | .669 | <.001 | <.001 | 342 |
| | 3 | 12,448 | .618 | .262 | <.001 | 104 |
| | 4 | 12,425 | .596 | .029 | <.001 | 19 |

*Note.* LCGA = Latent class growth analyses; BIC = Bayesian information criterion; LMR-LRT = Lo-Mendell-Rubin likelihood ratio test; BLRT = Bootstrap likelihood ratio test.

three-class model with stable conduct problems into a class with increasing and a class with decreasing conduct problems. To avoid neglecting distinguishable classes, we selected the four-class model for further analyses.

Fig 1 depicts the four trajectories of adolescents' conduct problems. Most adolescents fell into the stable low-risk group (73%), exhibiting no conduct problems before and during the pandemic (Table 3). The increasing low-risk group (9%) reported relatively few conduct problems before the pandemic, but these increased during the pandemic, shifting from the 'normal' to the 'borderline' range. The decreasing moderate-risk group (14%) showed moderate levels of pre-pandemic conduct problems which decreased during the pandemic and flattened over time. This group moved from the 'borderline' to the 'normal' range. Finally, a small group (4%) reported high levels of conduct problems before and during the pandemic and was considered the stable high-risk group, scoring in the 'abnormal' range.

Adolescents in the increasing low-risk group did not differ from those in the stable low-risk group, except for a higher likelihood of having a migration background. Compared to the stable low-risk group, adolescents in the decreasing moderate-risk and stable high-risk groups were more likely to be boys and perceived less family support. Additionally, adolescents in the decreasing moderate-risk group perceived less friend support than adolescents in the stable low-risk group.

## Hyperactivity-inattention problems trajectories

For hyperactivity-inattention problems, the quadratic growth model again showed a better fit than the linear growth model, $\Delta\chi^2 (4) = 42.91$, $p < .001$. The model fit improved with an increasing number of classes, as shown by decreasing BIC values and significant BLRT values ($p < .001$, Table 2). However, the six-class model did not improve the model according to the LMR-LRT ($p = .273$), and the decrease in BIC was relatively small. Therefore, the five-class model was selected for further analyses.

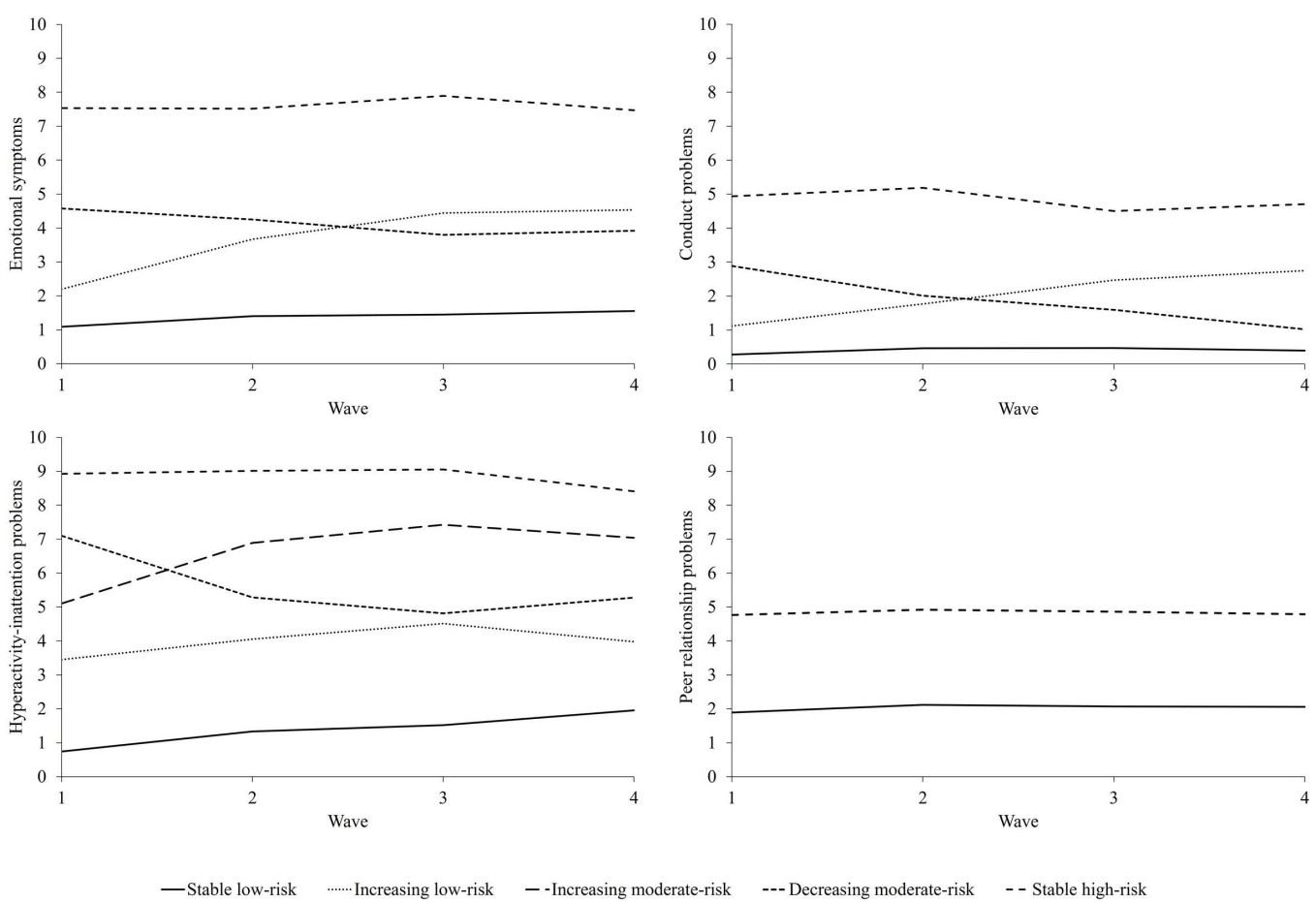

**Fig 1. Trajectories of mental health problems before and over the course of the COVID-19 Pandemic.**

Fig 1 depicts the five trajectories of adolescents' hyperactivity-inattention problems. The stable low-risk group (24%) reported minimal hyperactivity-inattention problems before the pandemic, which slightly increased during the pandemic (Table 3). The increasing low-risk group (31%) and increasing moderate-risk group (18%) had different levels of hyperactivity-inattention problems before the pandemic, but both showed an increase during the pandemic that leveled off over time. While the low-risk and increasing low-risk groups remained in the 'normal' range, the increasing moderate-risk group shifted from 'normal' to 'abnormal'. The decreasing moderate-risk group (8%) showed relatively high levels of pre-pandemic hyperactivity-inattention problems, which decreased during the pandemic and leveled off over time. This group moved from the 'abnormal' to the 'normal' range. Finally, the stable high-risk group (18%) reported high levels of hyperactivity-inattention problems before and during the pandemic with a slight decrease after the third wave, but remained in the 'abnormal' range.

Adolescents in the increasing low-risk and increasing moderate-risk groups were less likely to have a migration background and had higher family SES than adolescents in the stable low-risk group. Additionally, the increasing moderate-risk group perceived less family support than the stable low-risk group. Adolescents in the decreasing moderate-risk and stable low-risk groups were quite similar and only differed in terms of family SES, with the decreasing moderate-risk group originating from higher SES families. Compared to the stable low-risk group, adolescents in the high-risk group were less likely to have a migration background, had higher family SES, and perceived less family support.

**Table 3. Descriptives of the latent classes of the mental health indicators.**

| | Classes | Intercept | Linear slope | Quadratic slope | Gender (pro-portion boys) | Migration background (proportion non-Dutch) | Family SES (mean) | Family support (mean) | Friend support (mean) |
|---|---|---|---|---|---|---|---|---|---|
| Emotional symptoms | Stable low-risk (n=696) | 1.10*** (0.09) | 0.60* (0.23) | −0.19 (0.11) | 0.61 (0.03) | 0.32 (0.02) | 8.83 (0.11) | 6.25 (0.07) | 6.18 (0.07) |
| | Increasing low-risk (n=247) | 2.20*** (0.23) | 3.40*** (0.79) | −1.12** (0.33) | 0.32*** (0.05) | 0.22 (0.04) | 8.53 (0.22) | 6.01 (0.15) | 5.96 (0.15) |
| | Decreasing moderate-risk (n=367) | 4.59*** (0.20) | −1.03* (0.52) | 0.35 (0.24) | 0.32*** (0.04) | 0.22* (0.03) | 8.64 (0.16) | 5.47*** (0.13) | 5.58*** (0.12) |
| | Stable high-risk (n=209) | 7.51*** (0.14) | 0.44 (0.33) | −0.22 (0.17) | 0.15*** (0.03) | 0.18*** (0.03) | 8.32** (0.14) | 4.93*** (0.15) | 5.44*** (0.12) |
| Conduct problems | Stable low-risk (n=1109) | 0.28*** (0.04) | 0.39*** (0.11) | −0.17** (0.06) | 0.38 (0.02) | 0.23 (0.02) | 8.65 (0.07) | 6.00 (0.05) | 5.98 (0.05) |
| | Increasing low-risk (n=134) | 1.11*** (0.13) | 1.73** (0.60) | −0.45 (0.29) | 0.41 (0.05) | 0.35* (0.05) | 8.77 (0.28) | 5.87 (0.20) | 5.95 (0.19) |
| | Decreasing moderate-risk (n=214) | 2.88*** (0.08) | −1.81*** (0.37) | 0.44* (0.21) | 0.49* (0.04) | 0.23 (0.04) | 8.48 (0.13) | 5.17*** (0.13) | 5.51*** (0.11) |
| | Stable high-risk (n=63) | 4.97*** (0.40) | −0.15 (0.85) | −0.00 (0.45) | 0.59** (0.07) | 0.32 (0.07) | 8.87 (0.26) | 5.13** (0.33) | 5.53 (0.27) |
| Hyperactivity-inattention problems | Stable low-risk (n=365) | 0.76*** (0.09) | 1.14** (0.38) | −0.28 (0.19) | 0.37 (0.03) | 0.39 (0.03) | 8.24 (0.13) | 6.14 (0.10) | 5.92 (0.09) |
| | Increasing low-risk (n=466) | 3.44*** (0.18) | 1.72*** (0.44) | −0.72** (0.21) | 0.46 (0.03) | 0.24** (0.03) | 8.74* (0.14) | 5.88 (0.11) | 5.88 (0.10) |
| | Increasing moderate-risk (n=281) | 5.12*** (0.24) | 3.98*** (0.75) | −1.52*** (0.36) | 0.40 (0.04) | 0.20*** (0.04) | 8.79* (0.21) | 5.67** (0.16) | 6.06 (0.13) |
| | Decreasing moderate-risk (n=128) | 7.07*** (0.41) | −4.04** (1.23) | 1.59** (0.60) | 0.48 (0.07) | 0.28 (0.07) | 8.85* (0.25) | 6.06 (0.17) | 5.54 (0.18) |
| | Stable high-risk (n=281) | 8.92*** (0.11) | 0.42 (0.33) | −0.34* (0.16) | 0.35 (0.03) | 0.15*** (0.03) | 8.69* (0.16) | 5.20*** (0.14) | 5.90 (0.11) |
| Peer relationship problems | Stable low-risk (n=1179) | 1.90*** (0.08) | 0.39* (0.15) | −0.16* (0.08) | 0.42 (0.02) | 0.25 (0.01) | 8.73 (0.07) | 5.98 (0.05) | 6.12 (0.04) |
| | Stable high-risk (n=342) | 4.78*** (0.21) | 0.27 (0.34) | −0.13 (0.17) | 0.39 (0.03) | 0.26 (0.03) | 8.39* (0.14) | 5.35*** (0.11) | 5.18*** (0.11) |

*Note.* Stable low-risk class is the reference category. SES=Socioeconomic status.

*p<.05;

**p<.01;

***p<.001.

## Peer relationship problems trajectories

Lastly, the quadratic model for peer relationship problems also showed a better fit than the linear growth model, $\Delta\chi^2$ (1) = 6.34, $p=.012$. A two-class model showed improved model fit based on a lower BIC value and significant LMR-LRT and BLRT values ($p<.001$, Table 2). A three-class model did not improve the model according to a non-significant LMR-LRT ($p=.293$). Although a four-class model fitted the data significantly better than a three-class model, one class consisted of only 19 participants, which did not meet our criteria of $n>50$. Therefore, we selected the two-class model for further analyses.

Fig 1 depicts the two trajectories of adolescents' peer relationship problems. Most adolescents fell into the stable low-risk group (78%) and showed low levels of peer relationship problems before the pandemic but a slight increase during the pandemic which leveled off over time (Table 3). These scores remained in the 'normal' range. Adolescents in the stable high-risk group (22%) reported relatively high levels of peer relationship problems before and during the pandemic and were in the 'abnormal' range. No differences in gender and migration background were found between the two groups. However, compared to the stable low-risk group, adolescents in the stable high-risk group originated from lower SES families and perceived less family and friend support.

## Discussion

The current study investigated trajectories of adolescent mental health problems before and over the course of the COVID-19 pandemic. Emotional symptoms, conduct problems, and hyperactivity-inattention problems each exhibited four trajectories: a stable low-risk, increasing low-risk, decreasing moderate-risk, and stable high-risk trajectory. An additional increasing trajectory was identified for hyperactivity-inattention problems, where adolescents reported moderate levels before the pandemic and increases over the course of the pandemic. For peer relationship problems, only two trajectories emerged, a stable low-risk and a stable high-risk trajectory. Thus, consistent with our expectations, different groups of adolescents exhibited different trajectories of mental health problems when exposed to the pandemic. Across the indicators of mental health problems, adolescents in the stable high-risk groups perceived a relatively low level of family support and, sometimes, also a low level of friend support. Demographic characteristics (i.e., gender, migration background, family SES) were not systematically linked to specific trajectories of mental health problems.

Using the SDQ cutoff values based on normative data for Dutch 17-year-old adolescents [24], emotional symptoms in both the increasing-risk and decreasing-risk groups remained within the 'normal' range. This suggests that the COVID-19 pandemic did not have a substantial impact on adolescent emotional symptoms. Conduct problems may have been more affected by the pandemic, as the increasing-risk group shifted from the 'normal' to 'borderline' range, while the decreasing-risk group moved from 'borderline' to 'normal'. The pandemic seems to have had the largest impact on hyperactivity-inattention problems, as the most pronounced changes were observed for this subscale. The moderate increasing-risk group shifted from 'normal' to 'abnormal', while the decreasing-risk group moved from 'abnormal' to 'normal'.

### Trajectories of adolescent mental health problems

A relatively large number of adolescents were in the low-risk groups and reported minimal mental health problems before and during the pandemic, ranging from 24% for hyperactivity-inattention problems to 78% for peer relationship problems. Consistent with previous research, in the first 18 months after the outbreak, the pandemic seemed to have had minimal impact on the mental health of these adolescents. For example, Guzman Holst and colleagues [9] found that 62% to 84% of adolescents exhibited stable low levels of mental health problems during the pandemic. These findings suggest that many adolescents were able to adapt to the new circumstances and effectively manage the stress related to the pandemic, preventing them from developing mental health problems. Similar to our findings, research on potentially traumatic events indicates that the most common response to such events is resilience, which involves maintaining a high level of mental health from before to after the event [31]. In our study, resilience seemed to be the weakest for hyperactivity-inattention problems, possibly because COVID-19 measures such as remote learning and disruptions in daily routines may have affected concentration in particular.

A relatively small percentage of adolescents, ranging from 4% for conduct problems to 22% for peer relationship problems, were in the high-risk groups and had stable high levels of mental health problems before and during the pandemic. These findings indicate that adolescents who were vulnerable to mental health problems before the pandemic remained vulnerable during the pandemic. This is largely consistent with other studies suggesting that adolescents with pre-pandemic vulnerability are particularly susceptible to mental health problems during the pandemic [32,33]. However, our

findings suggest that the pandemic did not exacerbate mental health problems in these high-risk groups. Adolescents with pre-pandemic mental health problems may already have received treatment or support that may have helped them cope with the challenges of the pandemic. Also, the support provided by schools to at-risk adolescents may have prevented problems from worsening [34].

The increasing-risk groups consisted of adolescents who had few to moderate levels of mental health problems before the pandemic, but whose problems increased during the pandemic. Almost half of the adolescents reported increased hyperactivity-inattention problems, while 16% and 9% reported increased emotional symptoms and conduct problems, respectively. These increases in emotional symptoms and hyperactivity-inattention problems were most pronounced at the beginning of the pandemic and leveled off over time. Possibly, adolescents became more familiar with the new circumstances over time and may have mobilized cognitive and social resources to better deal with the challenges they faced [35]. Additionally, the increased attention to adolescent mental health during the pandemic may have helped to mitigate some of the potentially negative effects of COVID-19 measures. The large proportion of adolescents who showed an increase in hyperactivity-inattention problems may be explained by several factors, such as the shift to online learning environments [36], reduced teacher support [37], and decreased physical activity [38].

The decreasing-risk groups consisted of adolescents with moderate levels of pre-pandemic mental health problems which decreased over the course of the pandemic. Percentages ranged from 8% for hyperactivity-inattention problems to 24% for emotional symptoms. While emotional symptoms decreased modestly, conduct problems and hyperactivity-inattention problems decreased more substantially. The decrease in emotional symptoms for almost a quarter of the adolescents may be explained by the reduced stress of school and other social responsibilities due to COVID-19 measures such as school closings and stay-at-home directives [39]. The decreases in conduct problems and hyperactivity-inattention problems might be attributed to the increased time adolescents spent at home during lockdowns. From the perspective of Self-Determination Theory [40], the flexibility of remote learning may have allowed some adolescents to exercise greater autonomy. Some may also have experienced a greater sense of competence while working at home. Additionally, some adolescents may have experienced closer relationships with family members, enhancing their sense of relatedness. A greater satisfaction of the needs for autonomy, competence, and relatedness is linked to better well-being [40] and may be reflected in a decrease in conduct problems and hyperactivity-inattention problems. Moreover, reduced peer interactions may have prevented some adolescents from engaging in problem behaviors, such as fighting, and the home environment may have afforded fewer distractions than the classroom, leading to improved concentration.

### Predictors of adolescent mental health problems trajectories

We investigated whether family and friend support could help characterize the different trajectories. Our hypothesis that adolescents who received less family and friend support would be overrepresented in the stable high and increasing mental health problems trajectories is largely supported by our findings. Compared to the stable low-risk group, adolescents with stable high levels of emotional symptoms, conduct problems, hyperactivity-inattention problems, and peer relationship problems perceived less family support, and for emotional symptoms and peer relationship problems also less friend support. Additionally, adolescents with moderate, increasing levels of hyperactivity-inattention problems perceived less family support than the low-risk group, suggesting that these adolescents may lack the buffering effect of social support. In contrast, adolescents with moderate, decreasing levels of hyperactivity-inattention problems reported levels of family and friend support comparable to the stable low-risk group. Support from family and friends may thus have helped adolescents in this group cope with COVID-19 measures, for example by helping them maintain a daily routine, providing help with schoolwork, and reducing distraction compared to being at school.

Contrary to our hypothesis, the moderate decreasing-risk group of emotional symptoms and conduct problems perceived less family and friend support than the stable low-risk group. Additionally, there were no differences in perceived family and friend support between adolescents with stable low mental health problems and those with low but increasing

mental health problems during the pandemic. These findings suggest that social support assessed before the pandemic cannot explain changes in adolescent emotional symptoms and conduct problems during the pandemic. Family and friend support seemed to be more strongly associated with initial levels of emotional symptoms and conduct problems than with the course of mental health problems during the pandemic.

Furthermore, we investigated whether demographic variables were associated with the identified trajectories. Associations between migration background and family SES and the trajectories of the different mental health indicators were inconsistent. Adolescents with a migration background were overrepresented in the increasing-risk group of conduct problems and underrepresented in the decreasing-risk group of emotional symptoms. Contrary to our hypothesis, they were also underrepresented in the high-risk group of emotional symptoms, and in both the increasing-risk and high-risk groups of hyperactivity-inattention problems. Adolescents from lower SES families were overrepresented in the high-risk groups of emotional symptoms and peer relationship problems and underrepresented in the decreasing-risk group of hyperactivity-inattention problems. However, adolescents from lower SES families were not more vulnerable to experiencing an increase in mental health problems during the pandemic. They were even underrepresented in the increasing-risk and high-risk groups of hyperactivity-inattention problems. Previous studies among adolescents in the Netherlands also indicated that having a migration background is related to more conduct problems and fewer hyperactivity-inattention problems [41], while lower family SES is associated with more emotional symptoms and peer relationship problems, and fewer hyperactivity-inattention problems [42]. Thus, neither having a migration background nor coming from a lower SES family should be considered direct risk factors for developing or experiencing higher levels of mental health problems during the COVID-19 pandemic.

Adolescents with a migration background or from lower SES families are possibly too heterogeneous, with diverse experiences and circumstances that may more directly influence their mental health. Factors such as parental employment in essential sectors (e.g., healthcare) or living in crowded homes with limited opportunities for remote work or effective study can exacerbate stress within families, potentially impacting adolescent mental health. Moreover, adolescent subjective SES is more strongly associated with adolescent mental health problems than family SES and can even buffer the association between family SES and mental health problems [42]. Thus, factors such as parental employment type, living arrangements, and subjective SES may be better able to predict which adolescents experienced changes in mental health problems during the pandemic than migration background or family SES.

Furthermore, our results suggest that the COVID-19 pandemic has created unique conditions that temporarily leveled typical demographic or social disparities in mental health outcomes. For instance, widespread disruptions and shared experiences of uncertainty, social isolation, and restrictions could have overridden individual differences such as having a migration background or growing up in lower SES families, which are associated with greater vulnerability to mental health problems. This aligns with the idea that strong situations constrain behavioral options and limit variability among individuals [43]. In this sense, the COVID-19 pandemic could be considered a strong situation in which personal characteristics may have been less influential on mental health outcomes than in a context outside the pandemic.

Finally, girls were overrepresented in the increasing-risk, decreasing-risk, and high-risk groups of emotional symptoms, while boys were overrepresented in the decreasing-risk and high-risk groups of conduct problems. Previous research has shown that girls reported greater increases in internalizing problems than boys during the pandemic [18]. However, our findings suggest that gender did not necessarily drive vulnerability to emotional symptoms during the pandemic. This is evident from the overrepresentation of girls not only in the increasing-risk group but also in the group showing a decrease in emotional symptoms. Similarly, boys were overrepresented in the trajectories with the highest pre-pandemic levels of conduct problems, whether these problems remained stable or decreased during the pandemic. In line with existing research on gender differences in internalizing and externalizing mental health problems [17], these results suggest that gender is more strongly associated with pre-pandemic levels of emotional and conduct problems than with the trajectories of these problems during the pandemic.

**Limitations and future directions**

This study has the strengths of measuring mental health problems before and during the pandemic over 18 months in a large sample of adolescents. Nevertheless, its results should be considered in light of some limitations. First, while it is possible that the variations in mental health problems were related to the pandemic, adolescents experience different trajectories of mental health problems even in the absence of a pandemic [44,45]. This implies that increases and decreases in mental health problems may also reflect the typical course of mental health problems. Ideally, the findings of this study would be compared with those of a similar study conducted outside the context of the pandemic, but, as far as we know, such research is currently unavailable. Second, our sample consisted of students from tertiary vocational schools in one city in the Netherlands, which may limit the generalizability of our findings to other populations. For example, adolescents participating in this type of education may have been more vulnerable to being in the increasing mental health problems group. This could be because they are trained for jobs that require practical training and experience, and school closures may have had a greater impact on their training.

Third, the conduct problems and peer relationship problems subscales showed low internal consistency. Nevertheless, the SDQ has generally demonstrated adequate psychometric properties among Dutch and late adolescents [46,47], indicating that using the SDQ to assess mental health in the present sample is appropriate. Furthermore, we only measured family and friend support prior to the COVID-19 pandemic to investigate how the situation before the pandemic affects adolescent mental health trajectories. Further research could explore how social support developed during the pandemic and how this relates to the different trajectories.

## Conclusion

Given the extraordinary circumstances of the COVID-19 pandemic and its potential impact on adolescent mental health, it is important to examine how mental health problems have developed over the course of the pandemic and identify adolescents who are vulnerable to mental health problems. Our findings underline the diverse impact of the pandemic on mental health problems among adolescents. We found different trajectories for different indicators of mental health problems, encompassing stable low, stable high, increasing, and decreasing trajectories. The findings suggest that interventions and support strategies for adolescents to cope with stressful circumstances should be tailored to the specific needs of different groups of adolescents. Most adolescents reported few mental health problems, especially conduct and peer relationship problems. Adolescents with high levels of mental health problems before and during the pandemic reported relatively low levels of family support, and for some problems also low levels of friend support. We found little evidence that allowed us to systematically characterize subgroups of adolescents with increasing or decreasing mental health problems during the pandemic. This suggests that factors other than gender, migration background, family SES, and family and friend support may influence the course of mental health problems during the pandemic.

## Supporting information

**S1 Checklist. GRoLTS Checklist - Guidelines for reporting on latent trajectory studies.**
(DOCX)

**S2 Checklist. STROBE Statement - Checklist of items that should be included in reports of cohort studies.**
(DOCX)

## Acknowledgments

The authors thank the schools, teachers, and adolescents who participated in the YOUth Got Talent project.

## Author contributions

**Conceptualization:** Coriena de Heer, Catrin Finkenauer, Gonneke Stevens.

**Data curation:** Coriena de Heer.

**Formal analysis:** Coriena de Heer.

**Methodology:** Coriena de Heer.

**Supervision:** Catrin Finkenauer, Gonneke Stevens.

**Visualization:** Coriena de Heer.

**Writing – original draft:** Coriena de Heer.

**Writing – review & editing:** Coriena de Heer, Catrin Finkenauer, Gonneke Stevens.

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
