## [Decision Letter · Decision Letter 0]

25 Oct 2024

PMEN-D-24-00443

Different trajectories of adolescent mental health problems before and over the course of COVID-19: Evidence of increase, decrease, and stability

PLOS Mental Health

Dear Dr. de Heer,

Thank you for submitting your manuscript to PLOS Mental Health. After careful consideration, we feel that it has merit but does not fully meet PLOS Mental Health’s publication criteria as it currently stands. Therefore, we invite you to submit a revised version of the manuscript that addresses the points raised during the review process.

We look forward to receiving your revised manuscript.

Kind regards,

Gareth Hagger-Johnson

Academic Editor

PLOS Mental Health

Journal Requirements:

 1. Please amend your detailed Financial Disclosure statement. This is published with the article. It must therefore be completed in full sentences and contain the exact wording you wish to be published. **Please only choose the relevant sentences from below** 1. Please clarify all sources of funding (financial or material support) for your study. List the grants (with grant number) or organizations (with url) that supported your study, including funding received from your institution. 2. State the initials, alongside each funding source, of each author to receive each grant.3. State what role the funders took in the study. If the funders had no role in your study, please state: “The funders had no role in study design, data collection and analysis, decision to publish, or preparation of the manuscript.”4. If any authors received a salary from any of your funders, please state which authors and which funders. If you did not receive any funding for this study, please simply state: “The authors received no specific funding for this work.” 2. Thank you for uploading your study's underlying data set. Unfortunately, the repository you have noted in your Data Availability statement does not qualify as an acceptable data repository according to PLOS's standards.  At this time, please upload the minimal data set necessary to replicate your study's findings to a stable, public repository (such as figshare or Dryad) and provide us with the relevant URLs, DOIs, or accession numbers that may be used to access these data. For a list of recommended repositories and additional information on PLOS standards for data deposition, please see https://journals.plos.org/plosone/s/recommended-repositories. 

Additional Editor Comments (if provided):

Thank you for submitting your manuscript "Trajectories of adolescent mental health problems before and during the COVID-19 pandemic" to PLOS Mental Health. We have now received comments from two expert reviewers, and I am pleased to say that both find your study valuable and methodologically sound. They appreciate the large sample size, longitudinal design, and consideration of various demographic factors. However, they have identified several areas where revisions could enhance the manuscript's quality and impact.

Based on the reviewers' comments, I would like to request the following revisions:

Methodology and Reporting:

Please follow the GRoLTS-checklist guidelines for reporting latent trajectory studies, as suggested by Reviewer 1. This will ensure comprehensive reporting of your analytical choices.

Provide clearer justification for your analytical decisions, particularly regarding the latent class growth analysis.

Consider using a three-step approach for analyzing the impact of auxiliary predictors on class membership, as detailed by Reviewer 1. This method would account for classification uncertainty and potentially yield more robust results.

Results Interpretation:

Contextualize the observed changes in mental health scores with established severity thresholds for the SDQ. This will help readers understand the clinical significance of the identified trajectories.

Explore possible explanations for why some adolescents showed improved mental health during the pandemic, as suggested by Reviewer 2.

Discussion Enhancement:

Deepen the theoretical exploration of your findings, particularly regarding gender differences and the lack of consistent associations with demographic characteristics.

If applicable, consider discussing the functional impact supplement of the SDQ.

Minor Clarifications:

Provide more precise definitions for certain terms (e.g., "migration background") and age ranges.

Strengthen the introduction by emphasizing the importance of long-term trajectories and distinguishing between peer socialization and formal mental health services.

Both reviewers believe your manuscript has the potential for publication in PLOS Mental Health after addressing these concerns. Please carefully consider all suggestions, particularly those related to methodology and result interpretation, to strengthen your paper's contribution to the field.

We look forward to receiving your revised manuscript. Please include a point-by-point response to the reviewers' comments along with your revisions.

Sincerely,

Dr. Gareth Hagger-Johnson

Reviewers' comments:

Reviewer's Responses to Questions

**Comments to the Author**

1. Does this manuscript meet PLOS Mental Health’s publication criteria ? Is the manuscript technically sound, and do the data support the conclusions? The manuscript must describe methodologically and ethically rigorous research with conclusions that are appropriately drawn based on the data presented.

Reviewer #1: Yes

Reviewer #2: Yes

2. Has the statistical analysis been performed appropriately and rigorously?

Reviewer #1: Yes

Reviewer #2: N/A

3. Have the authors made all data underlying the findings in their manuscript fully available (please refer to the Data Availability Statement at the start of the manuscript PDF file)?

Reviewer #1: Yes

Reviewer #2: Yes

4. Is the manuscript presented in an intelligible fashion and written in standard English?

Reviewer #1: Yes

Reviewer #2: Yes

5. Review Comments to the Author

Reviewer #1: Overall, the manuscript presents a methodologically robust study with sound findings. The conclusions are well-framed and supported by the data, with appropriate consideration of the study's limitations and ethical context. Further theoretical exploration could enhance the interpretation, but the research offers valuable insights into the complex mental health trajectories of adolescents during the COVID-19 pandemic. The statistical analysis is rigorous, effectively capturing changes in adolescent mental health over time. The large sample size and consideration of demographic factors provide a strong foundation for the conclusions, despite some limitations in generalisability.

The authors have made the dataset available as stated: "The dataset and syntaxes are available in the OSF database (https://osf.io/j2t6w/). We will make the OSF folder public after the manuscript is accepted."

Minor Revisions:

Introduction: When discussing the importance of peer and formal support, it would be beneficial to differentiate between peer socialisation and formal mental health services. This distinction could enrich the argument that both were disrupted during the pandemic, e.g., "Both peer socialisation and formal mental health services, vital for adolescent well-being, were significantly disrupted due to school closures and social distancing measures."

Introduction: While the section on mental health trajectories is thorough, emphasising the importance of examining long-term trajectories would strengthen the argument. A brief note on how understanding these trajectories can inform interventions more effectively than short-term studies could be added.

Methods: The age range at T1 is presented as an average with a standard deviation, but specifying the actual age range (e.g., 15-21) would clarify the group being studied.

Methods: A clearer definition of "migration background" would enhance clarity, such as: "Participants were considered to have a migration background if at least one parent was born outside the Netherlands."

Methods: Providing a brief rationale for selecting latent class growth analysis (LCGA) could add depth.

Discussion: While the discussion acknowledges that some adolescents exhibited resilience while others struggled, exploring why some experienced decreases in conduct problems or hyperactivity-inattention during the pandemic could strengthen the conceptual implications. Linking this to developmental or psychological theories (e.g., the stress-buffering model of social support) would deepen the analysis.

Discussion: Further exploration of why girls were overrepresented across multiple trajectories (e.g., increasing, decreasing, and high-risk) could add nuance. Exploring how gender roles or coping mechanisms may have influenced these results would provide valuable insight.

Discussion: The inconsistent associations between demographic characteristics (e.g., migration background, SES) and trajectories warrant further exploration. While the findings that these variables do not systematically predict mental health outcomes are noted, a more nuanced discussion could elucidate why this might be the case, considering differences in school closures, family dynamics, or cultural factors.

Discussion: While the discussion highlights some non-significant findings (e.g., no systematic link between demographic factors and mental health trajectories), it could further explore why expected patterns, such as migration background or SES predicting vulnerability, did not emerge. Considering whether the pandemic created conditions that levelled typical social or demographic disparities would add depth to the interpretation.

Reviewer #2: PMEN-D-24-00443

SUMMARY

The article examines the trajectories of mental health problems in adolescents before and during the COVID-19 pandemic. Using data from 1,522 adolescents collected at four different points between 2019 and 2021, the study identifies various trajectories of emotional symptoms, conduct problems, hyperactivity, and peer relationship issues. Four main trajectories were identified for emotional symptoms, conduct problems, and hyperactivity: stable low risk, stable high risk, increasing low risk, and decreasing moderate risk. For ADHD, 5th trajectory was found: increasing moderate risk. For peer relationship problems, two trajectories were found: stable low and stable high.

The findings show that the pandemic did not affect all adolescents in the same way. Some experienced increasing symptoms (especially those with low starting values), others saw decreases, while the majority remained stable (at both a high and low level). Adolescents with high levels of mental health problems before the pandemic also reported relatively low levels of family and peer support. Demographic characteristics such as gender, migration background and socioeconomic status were not consistently linked to specific trajectories. Overall, these results emphasize the need for tailored intervention strategies to meet the specific needs of different groups of adolescents during stressful times.

GENERAL COMMENT

The subject is very interesting, and the authors have partly dealt with it in the appropriate way. Of note is also the high quality of the data. This article is therefore of interest to the psychopathology research community as well as to the public health research community that might compose the broad readership of Plos Mental Health.

The introduction is clear and well-documented, explaining both the challenges and the strengths of the study. On the latter point, the authors have a larger sample size than other previous studies, measures of mental health that have been widely validated internationally (internalizing/externalizing problems) and collected at different points in time (before, during and after the pandemic), and socio-demographic data that are relevant for studying inter-individual variability within classes.

The objectives and hypotheses are also set out very clearly.

I found the discussion objective, avoiding over-interpretation.

Finally, the statistical framework used by the authors is appropriate for meeting their first objective (Identifying different trajectories of adolescent mental health problems before and during the COVID-19 pandemic, using a sizeable, prospective, longitudinal study).

However, I have a major concern with the way the authors described their analytic choices (which often lack justification), and also with the method they chose to achieve their second objective (exploring whether migration background, family SES, gender, family support, and friend support were associated with the identified trajectories). I think the authors have every opportunity to address these concerns, and I hope I can help them in this endeavor.

Overall, I think that the methodological choices made by the authors to study the time series must be better justified. I also think that the comprehension of the latent class model’s results on the light of auxiliary predictors could greatly benefit from an alternative approach.

To summarize, this article has the potential to be published in Plos Mental Health. I hope that the authors will find in my few comments an opportunity to improve the quality of their manuscript.

MAJOR

1. I would recommend the authors to follow the guidelines for reporting on latent trajectories study by van de Schoot et al (van de Schoot R, Sijbrandij M, Winter SD, Depaoli S, Vermunt JK. 2016 The GRoLTS-checklist: guidelines for reporting on latent trajectory studies. Struct Equat Model 24, 451-467, doi:10.1080/10705511.2016.1247646). The purpose of this paper is to present criteria that should be included when reporting the results of latent trajectory analysis across research fields. It summarizes in a very comprehensive way several important decisions that researchers must make while writing the results of their study (for example, why choosing a LGCA model instead of a LGMM model?).

I would ask authors to check carefully whether they complete all the points that might concern them, and to include in the revised version of the manuscript any changes (analytic, semantic, graphical, etc) that might result from this examination.

2. Related to this, I think that the comprehension of the latent class model’s results on the light of auxiliary predictors could greatly benefit from a more elaborated methodology. The authors propose to study whether gender, migration background, family support, friend support, and family socio-economic status, have an impact on the distribution of subjects between the estimated classes. To do this, they look at each class separately (eg, increasing low-risk) to see if subjects differ in each of the factors listed above.

I see some limitations to this approach. First, there is the use of multiple chi-square tests to understand the structure of each class whose significance threshold, as far as I understand, is not adjusted for this multiple testing. Second, beyond the fact that this approach hampers the overall understanding of the results, its main problem is that it treats classes as (real) observed objects, whereas they are merely estimated objects involving subjects classified with more or less statistical precision. Clark & Muthen explain this problem very clearly in their methodological article « Relating Latent Class Analysis Results to Variables not Included in the Analysis » (http://www.statmodel.com/download/relatinglca.pdf):

« Suppose a 2-class model and take two individuals, one with a probability of 1.0 for belonging to Class 1 and 0.0 for Class 2 and the other with a probability of 0.51 for belonging to Class 1 and 0.49 for Class 2. Both individuals would be assigned and treated as members of Class 1 in the subsequent analyses. But the analyses does not take into account that the two individuals have different probabilities of being in the same class and instead are treated as if they both have a probability of 1.0 of being in Class 1. This will distort estimates because individuals are forced into their most likely latent classes. The standard errors will also be incorrect because the analysis does not take into account the uncertainty of the classification but treats it as an observed variable. This poses a problem because incorrect standard errors can lead to erroneous conclusions about the significance of an effect. »

These difficulties can be mitigated by using a three-step approach, again well described in the van de Schoot paper. I would like to encourage the authors to apply such an approach. The 1st step determines the number of latent classes without the predictors on class membership.

The 2nd step simply consists in saving the most likely class membership and related probability (classification precision), and then merged it with the original data. The 3rd step consists in analysing class membership separately from the latent trajectory model using a (univariate and then backward stepwise) multinomial regression analysis weighted for the posterior probability of class membership (hence accounting for classification imprecision). The authors may eventually dig even further by investigating more precisely how the probability of being assigned into a given class of symptoms trajectory varies for each level of a predictor of interest (eg, gender, family SES).

A practical application of this approach can be found in:

Edjolo A, Dartigues JF, Pérès K, Proust-Lima C. (2020). Heterogeneous long-term trajectories of dependency in older adults: the PAQUID cohort, a population-based study over 22 years. J. Gerontol Med Sci 75, 2396-2403. (doi:10.1093/gerona/glaa057)

or in:

Safra, L., Lettinga, N., Jacquet, P. O., & Chevallier, C. (2022). Variability in repeated economic games: comparing trust game decisions to other social trust measures. R Soc Open Sci, 9(9), 210213 (https://doi.org/10.1098/rsos.210213)

3. The authors report that a significant proportion of teens saw their problems increase or decrease during the pandemic. It would be interesting to put the scores observed for the “increasing“ and the “decreasing“ classes of subjects at the end of the pandemic into perspective with the severity thresholds identified by epidemiological studies on SDQ. For example, if we refer to the reference studies used by SDQ designers to establish severity cut-offs (https://www.sdqinfo.org/a0.html), it appears that the scores of teens belonging to the “increasing” groups remain within non-problematic limits. The only “increasing“ score that appears to be “on the edge“ is the Emotional symptoms score. Considering this, a question can be addressed: To what extent can the “changes” identified by the latent class model be considered as genuine changes rather than random fluctuations around the mean? This possibility is already raised by the authors in the discussion, but it might be interesting to elaborate more on it.

MINOR

4. The SDQ also has a functional impact supplement. Was this supplement administered, and if yes, why was it not the object of temporal analyses?

6. PLOS authors have the option to publish the peer review history of their article (what does this mean? ). If published, this will include your full peer review and any attached files.

**Do you want your identity to be public for this peer review?** For information about this choice, including consent withdrawal, please see our Privacy Policy .

Reviewer #1: No

Reviewer #2: No

---

## [Editor Report · Decision Letter 1]

29 Jan 2025

PMEN-D-24-00443R1

Different trajectories of adolescent mental health problems before and over the course of COVID-19: Evidence of increase, decrease, and stability

PLOS Mental Health

Dear Dr. de Heer,

Thank you for submitting your manuscript to PLOS Mental Health. After careful consideration, we feel that it has merit but does not fully meet PLOS Mental Health’s publication criteria as it currently stands. Therefore, we invite you to submit a revised version of the manuscript that addresses the points raised during the review process.

We look forward to receiving your revised manuscript.

Kind regards,

Gareth Hagger-Johnson

Academic Editor

PLOS Mental Health

Additional Editor Comments (if provided):

In addition to the requests made by both reviewers, please can you convince readers that latent trajectories aren't statistical artefacts driven by regression to the mean. In any longitudinal dataset, there will always be a group starting high and decreasing. Are these really latent classes?

Please include both a GRoLTS-Checklist and a STROBE checklist ensuring you include as much detail from both allowing readers to critically appraise the paper in a structured way.

Regards,

Dr. Gareth Hagger-Johnson
---

## [Editor Report · Decision Letter 2]

14 Mar 2025

Different trajectories of adolescent mental health problems before and over the course of COVID-19: Evidence of increase, decrease, and stability

PMEN-D-24-00443R2

Dear de Heer,

We are pleased to inform you that your manuscript 'Different trajectories of adolescent mental health problems before and over the course of COVID-19: Evidence of increase, decrease, and stability' has been provisionally accepted for publication in PLOS Mental Health.

Thank you for a detailed and clear revision, responding to reviewer and editor comments.

On the STROBE checklist, you need to add the page number when the article is published and the financial statement is included. The journal office will liaise with you.

Best regards,

Gareth Hagger-Johnson

Academic Editor

PLOS Mental Health